# GeRaF: Neural Geometry Reconstruction from Radio Frequency Signals

**Jiachen Lu**[*], **Hailan Shanbhag**[*], **Haitham Al Hassanieh**
École Polytechnique Fédérale de Lausanne (EPFL)

## Abstract

GeRaF is the first method to use neural implicit learning for near-range 3D geometry reconstruction from radio frequency (RF) signals. Unlike RGB or LiDAR-based methods, RF sensing can see through occlusion but suffers from low resolution and noise due to its *lensless* imaging nature. While lenses in RGB imaging constrain sampling to 1D rays, RF signals propagate through the entire space, introducing significant noise and leading to cubic complexity in volumetric rendering. Moreover, RF signals interact with surfaces via specular reflections, requiring fundamentally different modeling. To address these challenges, GeRaF (1) introduces filter-based rendering to suppress irrelevant signals, (2) implements a physics-based RF volumetric rendering pipeline, and (3) proposes a novel lensless sampling and lensless alpha blending strategy that makes full-space sampling feasible during training. By learning signed distance functions, reflectiveness, and signal power through MLPs and trainable parameters, GeRaF takes the first step towards reconstructing millimeter-level geometry from RF signals in real-world settings.

## 1 Introduction

Geometry reconstruction is a fundamental problem that enables a wide range of applications in fields such as virtual reality [14, 21, 32] and robotics [22, 29]. In recent years, neural reconstruction methods [55, 51, 40, 41, 35] have gained significant attention. A key advantage of these methods is their ability to represent the geometry of a scene continuously, which can help in many downstream tasks. However, vision-based sensors often struggle in environments with adverse weather conditions [67, 20, 37, 34] or even become completely unusable when objects are obscured by occlusions [2, 65, 66, 30, 57, 12, 13].

In contrast, radio frequency (RF) sensing, specifically wireless millimeter-wave (mmWave) sensing, has the ability to see *through* occlusions and remains robust under challenging visibility conditions and, unlike X-Ray, is not dangerous to humans [56], making it a compelling alternative for 3D reconstruction. This could open up a plethora of applications; for example, seeing whether items inside a box are the correct items, are damaged, or pose a threat, without having to ever open up the box. On the other hand, mmWave resolution is extremely low compared to vision. Fig. 1 shows an example of one radar image, which shows significantly less visual context than camera images provide, making it hard to directly extract 3D reconstruction from a radar image alone. In order to perform more complete 3D reconstruction, some works [2, 65, 30, 57, 13] have done 3D point cloud reconstruction or human body tracking by leveraging movement or emulating larger antenna apertures, however, the recovered point clouds from RF are still too sparse and noisy to reliably recover detailed geometry. Other works [6, 24] have proposed using neural implicit representations for RF sensing, aiming to concentrate scene information from heavy noise through neural optimization.

---

[*]Co-primary first authors, indicates equal contribution.

39th Conference on Neural Information Processing Systems (NeurIPS 2025).

However, their systems are designed for large-scale environments and are far from achieving millimeter-level precision. Compared to near-field surface reconstruction, propagation characteristics of RF signals change dramatically when objects are close to the radar [57, 45], meaning processing methods for large scale scenes, such as beamforming [5], introduces a considerable amount of distortion and noise, often exceeding the noise acceptable for gradient descent leading to unstable optimization.

In this paper, we propose ***the first method*** to overcome the fundamental dilemma between massive RF noise in the input and millimeter-level geometry reconstruction by using neural representation learning, shown in Fig. 1. Achieving this goal, however, is not so straightforward. Using RF imaging directly to infer surface geometry presents fundamentally different challenges from traditional image-based approaches, in three main ways. **(1)** A key challenge is the absence of a lens-based imaging model in wireless systems. While vision rendering uses lenses to filter and focus relevant light rays, wireless systems operate via *lensless imaging*, capturing all incoming RF signals without directional fil-

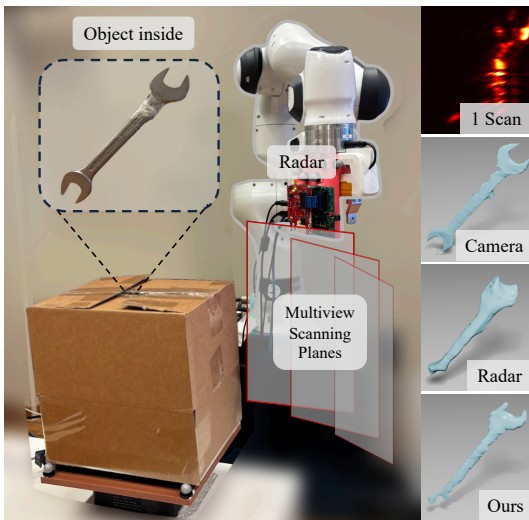

Figure 1: GeRaF is the first method to reconstruct millimeter-level geometry of non-line-of-sight objects using radio frequency (RF) signals by using neural implicit representations. ***Right***, shows the radar heatmap from one scan, a camera scan, the surface directly reconstructed from the radar heatmap images, and surface output from GeRaF.

tering. This leads to low signal-to-noise ratios, as irrelevant signals cannot be excluded. **(2)** There is a fundamental difference in the propagation characteristics between RF signals and visible light. While visible light propagation is predominantly determined by scattering, RF reflections are dominated by specular reflections. Therefore, volumetric rendering techniques that are designed for scattering-based vision sensors are unsuitable for RF imaging. **(3)** Finally, a major challenge is the high computational cost of lensless volumetric rendering. Unlike lens-based rendering, which samples along 1D rays, lensless rendering requires sampling across the full 3D space, leading to cubic complexity, making naïve implementations intractable.

To address the outlined challenges, we propose the following key contributions: **(1)** We perform geometry reconstruction by introducing a **matched filter (MF)** into the framework that discards irrelevant signals and preserves the most informative ones. **(2)** We implement **physics-based RF volumetric rendering**, which integrates a RF-specific reflection model and incorporates a mathematical framework for specular reflections directly into the rendering process. **(3)** We propose a novel **lensless sampling and lensless alpha blending** strategy that enables comprehensive scene coverage while significantly reducing computational cost, making volumetric rendering feasible for millimeter-level RF applications. Building upon these components, we design our neural model with three key sub-networks: a Signed Distance Function Network, a Reflective Network, and a Signal Power Prediction, which are implemented with MLPs or trainable parameters.

We evaluate our system using a 77 GHz mmWave radar mounted on a robotic arm, scanning a variety of real-world objects. Our results demonstrate that GeRaF takes a significant first step toward accurate 3D reconstruction from RF signals, outperforming currently adopted methods in both reconstruction quality and robustness to experimental noise. These findings highlight GeRaF's ability to infer more detailed scene geometry, as compared to current methods, even under challenging sensing conditions.

## 2  Related Work

**Vision-Based Multi-View 3D Reconstruction and Inverse Rendering** Traditional methods exploit photometric consistency across images [17] and fuse the resulting depth maps into a dense point cloud [16, 63, 58]. To obtain a surface representation, techniques such as Alpha Shapes [15] and Poisson Surface Reconstruction [27] are commonly applied as post-processing steps on the point cloud. With the rise of deep learning, methods like Neural Radiance Fields (NeRF) [39] and Gaussian

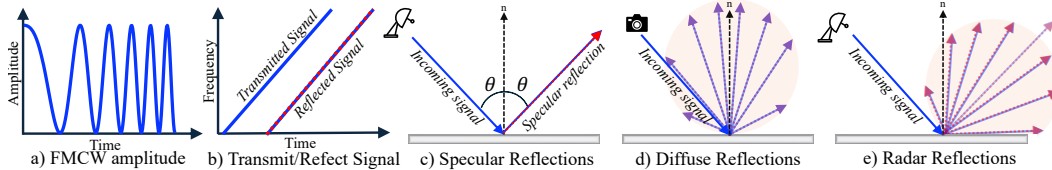

a) FMCW amplitude    b) Transmit/Refect Signal    c) Specular Reflections    d) Diffuse Reflections    e) Radar Reflections

Figure 2: (a) Frequency Modulated Continuous Waveform, (b) the reflected signal is a delayed version of the transmitted signal, to resolve distance. (c) Specular reflections dominate radar wave propagation vs (d) diffused reflections that dominate vision, (e) the radar reflections modeled to incorporate specular reflections and signal spread.

Splatting (3DGS) [28] use learnable parameters to reconstruct 3D scenes from multi-view images. Building upon neural implicit representations [42, 62, 61, 55] and 3DGS [23, 25, 33, 18, 8], recent work further decomposes scenes into explicit geometry and reconstruct surface meshes. Moreover, [46, 7, 41, 59, 26, 35, 19, 60] address the challenge of predicting lighting and material properties by considering complex light interactions, based on the explicit forward rendering equation with BRDF, a process known as inverse rendering. This technique is more physically grounded and brings reconstructions closer to real-world appearance. While these methods typically operate on lens-based RGB images, [38] proposes novel view synthesis on lensless RGB images. However, lensless RGB images can still be mapped to a lens-based imaging model without altering the underlying ray physics, unlike RF signals, where such transformation is fundamentally invalid. To date, all of these 3D representations are derived from *RGB images*, which are not easily transferable to *mmWave data*.

**3D Radar Imaging** Some works have used low frequency radars to estimate the 3D pose of humans and track them through walls and occlusions [2, 1, 65, 30]. However, these works leverage human motion to combat specularity, and are geared towards human mesh reconstruction. Past work also leverages deep learning in the context of millimeter wave radar data for imaging or point cloud completion, however these works primarily focus on the context of self-driving cars, where image resolution is not important and may not generalize to other objects [20, 47, 31, 48, 49].

More recently, [36, 68, 9] take inspiration from NeRF [39] to solve a different problem of estimating the wireless channel based on the predicted signal emitted from different reflectors in space. A group of work [4, 52] applies NeRF to simulated satellite images; [6, 24] tries to reconstruct 3D scenes by representing the wireless signal in the frequency domain, and learning the reflectance and occupancy of different points. However, all of these prior works try to perform radar image rendering tailored for reconstructing large scale scenes such as streets or satellite images and don't address close range high resolution object reconstruction, which requires different wave propagation modeling as explained in Appendix A.3. Authors of [50], proposes a method for 3D neural reconstruction of objects. However, their evaluation is limited to simulated data, which when dealing with wireless signals, cannot come close to representing the complexity of real world experiments.

It is important distinction to make, that all prior works of 3D neural reconstruction using radio frequency are either limited to simulated data or reconstruction for far range scenes, whereas our work is the first paper to propose complete high resolution 3D mesh reconstruction.

## 3 Technical Background

**Radar Basics** A mmWave radar works by transmitting a wireless signal and receiving back reflections that come from reflectors in the scene. It operates in the millimeter-wavelength frequency bands, and uses Frequency Modulated Continuous Wave (FMCW) and antenna arrays to help resolve spatial ambiguity. As shown in Fig. 2(a), the transmitted FMCW is a function that is linearly increasing in frequency over time. Considering a single reflector and a single antenna, the received chirp is simply a time-delayed version of the transmitted chirp, see Fig. 2(b). To resolve range ambiguity, the received chirp is multiplied with the conjugate of the transmitted chirp and can be expressed as a complex function:

$$s(t) = A \cdot e^{-j2\pi(f+kt)d/c} = A \cdot e^{-(j2\pi k\tau)t} \cdot e^{-j2\pi f\tau} \tag{1}$$

where $A$ is the signal amplitude, $d$ is the round-trip propagation distance, $c$ is the speed of light, $\tau = d/c$ is the round-trip delay, $f$ is the starting frequency, and $k$ is the chirp slope. For multiple reflectors, we simply receive the linear combination of all the reflections.

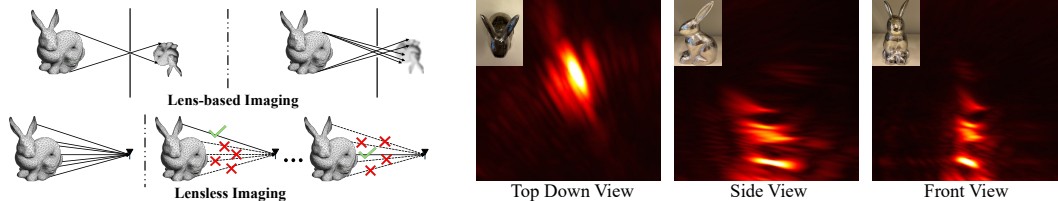

| (a) Lens-based vs lensless imaging | (b) Example MF image from measured radar data. |

Figure 3: a) Comparison between lens-based imaging and lensless imaging models. b) Matched filter image of the metal bunny, summed along the vertical, horizontal, and depth axis.

**Reflections** Unlike light, whose wavelength (order of $10^{-7}$ meters) is much smaller than surface irregularities, resulting in predominantly diffused reflection, RF signals have much longer wavelengths (order of $10^{-3}$ meters). In this case, the majority of surfaces appear relatively smooth to the RF signal, and specular reflections becomes more prominent than diffused scattering. For specular reflections, the angle of incidence is the same as the angle of reflection, while diffused reflections tend to scatter at different angles. The difference is illustrated in Fig. 2(c,d).

In this paper, we primarily expect specular reflections. We follow a shifted Lambertian model [44, 43], where the power of the reflections that return to the receiver, is scaled according to the normal and the signals reflection angle, see Fig. 2(e). The received power is modulated by the directional scaling factor $(\omega_o \cdot \omega_r)$, where $(\omega_o$ is the ideal specular vector and $\omega_r$ is the vector from the reflecting surface to the receiver (see Appendix A.2). The reflection strength also depends on material properties and the thickness of the object, which we model using a reflective coefficient $a$.

An additional power decay factor of $(4\pi u)^2$ is divided from the received reflection amplitude ($(4\pi u)^4$ for power), where $u$ is the distance between the point and antennas, to accurately model the free space path loss (see Appendix A.1). Given an input signal with amplitude $A_{\text{tx}}$, the received signal amplitude $A_{\text{rx}}$ can be expressed as:

$$A_{\text{rx}} \propto \frac{a}{(4\pi u)^2} A_{\text{tx}} \left(\omega_o \cdot \omega_r\right) \tag{2}$$

**Lensless imaging** Unlike traditional visual imaging systems that rely on lenses, radar imaging is inherently a **lensless** imaging technique. As illustrated in Fig. 3a, a pinhole in visual systems acts as a physical filter, allowing only one ray per scene point to reach the CMOS sensor while blocking unrelated rays. In photography, increasing the aperture size allows more light rays to enter, but this also leads to blur and noise in the image. As the aperture approaches an infinite size, the system transitions into a lensless imaging model. Lensless imaging, such as with radar antenna arrays, does not employ any physical pinhole—*all* incoming rays are received by all antennas.

In contrast, radar imaging relies on algorithmic processing to reconstruct the scene. The core idea involves applying a digital filter to remove unrealistic signals and keep the most relevant ones, leveraging the time-varying and phase-delay information of the received signals. Each received signal, arriving via a different path, carries a distinct phase shift defined by its propagation delay. The algorithm used for this process is called the **matched filter**.

**Differentiable Matched Filter** This algorithm is implemented by correlating the received signal with an ideal reference signal. From Eq. 1, the reference signal for a single reflector and a single antenna is known. For multiple antenna measurements, we have:

$$P(\mathbf{x}) = \left\|\sum_{i=1}^{N_{\text{ant}}} \sum_{t} s(i,t)\, e^{j2\pi k \tau_i t}\, e^{j2\pi f \tau_i}\right\|, \tag{3}$$

where $N_{\text{ant}}$ is the number of antennas, $\tau_i$ represent the round-trip delay corresponding to the $i$-th antenna, and $s(i,t)$ denotes the received signal at time $t$ from the $i$-th antenna. We omit the transmitted amplitude $A_{\text{tx}}$ since it is constant. This process effectively averages out signals that are inconsistent with the hypothesized signal model, a matched filter (MF) power image is shown in Fig. 3b.

To enable integration into a differentiable rendering framework, we also compute the backpropagation of MF. Both the forward and backward passes are implemented in parallel on the GPU. All implementation details are provided in Appendix A.4 and A.5.

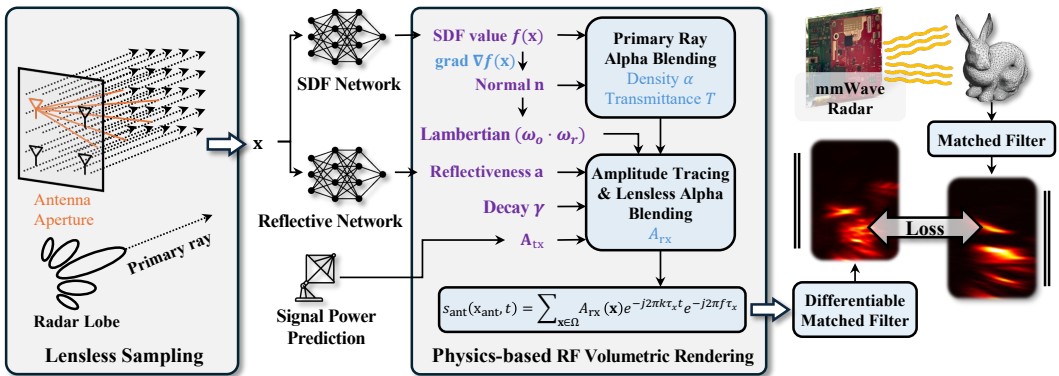

Figure 4: Overview of the GeRaF framework. (1) Lensless sampling replaces ray-based methods. (2) A neural implicit model predicts geometry, reflectivity, and power. (3) RF volumetric rendering simulates physical signal propagation. (4) Matched filtering produces MF power images (heatmaps). (5) An L2 loss compares the rendered and ground truth power for end-to-end training.

# 4 Overview

Given raw RF signals and the positions of all antennas across multiple antenna planes, GeRaF aims to reconstruct the surface geometry of the object by predicting a signed distance function (SDF), without requiring any additional information. GeRaF consists of five main components, illustrated in Fig. 4: (1) We begin with point sampling. Unlike the ray-based sampling used in vision-based methods [39], we propose a **lensless sampling** strategy that aligns with the physical nature of radar sensing. Details are provided in Sec. 6. (2) The sampled points are passed through three sub-networks to predict the scene's geometry, reflective properties, and radar signal power. (3) Using these predictions, along with surface normals (computed as gradients of the SDF), reflection directions, and power decay, we perform **physics-based RF volumetric rendering** for each antenna, as described in Sec. 5. To correctly compute opacity and transmittance under the lensless setting, we introduce a novel **lensless alpha blending strategy**, detailed in Sec. 6. (4) Next, we apply a matched filter (MF), defined in Eq. E.2, to compute the reflected power at each sampled point and generate the rendered MF heatmap. This step converts high-frequency time-domain signals into a lower-dimensional spatial representation of the scene. (5) Finally, we apply an L2 loss to minimize the discrepancy between the rendered heatmap and the ground truth, enabling end-to-end optimization of the entire framework via backpropagation.

# 5 Physics-based RF Volumetric Rendering

We begin this section with a simple ray-tracing-based sampling strategy. Assume we sample a single ray emitted from a receiver antenna located at $\mathbf{x_{ant}}$ in the direction $\omega_r$. A point $\mathbf{x}$ along the ray can then be represented as $\mathbf{x} = \mathbf{x_{ant}} + u\mathbf{r}$, where $u$ is the sampling distance along the ray, and $\mathbf{r}$ is the unit direction vector opposite to $\omega_r$. We first present *Signal Tracing*, which computes the frequency and phase shift of the signal. Then, we introduce *Signal Amplitude Rendering*, which "renders" the amplitude term of the received signal.

## 5.1 Signal Tracing

The Signal Tracing generates expected signal responses for all possible propagation paths based on a hypothesized scene or spatial configuration. Simulating the received signal corresponds to modeling the received chirp at each individual antenna. Using the single-reflector expression in Eq. 1, the reference signal for a scene with multiple reflectors is the superposition of contributions from all reflectors. However, in practice, there are no explicit "reflectors"; instead, we are dealing with a continuous 3D scene. We treat every point $\mathbf{x}$ in the computational space $\Omega_{\text{pts}}$ as a potential reflector, regardless of whether it lies in free space or on the surface of an object. Instead, volume density and reflective properties are encoded in the amplitude term $A(\mathbf{x})$. The received signal is expressed in Eq. 4 and is implemented in parallel on the GPU. Implementation details are provided in

Appendix A.4 and A.5..

$$s(\mathbf{x_{ant}}, t) = \sum_{\mathbf{x} \in \Omega_{\mathrm{pts}}} A_{\mathrm{rx}}(\mathbf{x}) \, e^{-j2\pi k \tau_{\mathbf{x}} t} \, e^{-j2\pi f \tau_{\mathbf{x}}}, \tag{4}$$

## 5.2 Signal Amplitude Tracing

The amplitude of the reflected radio frequency signal is influenced by several factors, including surface geometry, material reflectivity, reflection angle, and power decay due to propagation distance. Unlike vision-based NeRF rendering [39], radar sensing involves a **round-trip** path, requiring us to model both the incoming ray from the transmitter and the outgoing ray to the receiver.

Given the opaque density [55] $\rho(u)$, the *accumulated transmittance* along the incoming path can be computed as $T(u) = \exp\left(- \int_0^u \rho(v) \, \mathrm{d}v\right)$. However, due to the round-trip path of radar signals, the effective accumulated transmittance becomes $T(u)^2$, meaning that only a fraction $T(u)^2$ of the signal remains after returning to the receiver.

By combining volumetric rendering, reflectivity, the reflection angle and power decay as defined in Eq. 2, the amplitude of the reflected signal at the receiver antenna (rx), assuming a symmetric return path along the same ray, is given by:

$$A_{\mathrm{rx}}(\mathbf{x}_{\mathrm{ant}}, \omega_r) = \mathbf{a}(u) \, (\omega_o \cdot \omega_r) \, \gamma(u) \, T(u)^2 \, \rho(u) \, \mathbf{A}'_{\mathrm{tx}} \, \mathrm{d}t, \tag{5}$$

where $\mathbf{a}(u)$ is the reflectivity at distance $u$, $\omega_o$ is the outgoing direction computed as $\omega_o = \omega_i - 2(\mathbf{n} \cdot \omega_i)\mathbf{n}$, with $\omega_i$ being the incoming direction and $\mathbf{n}$ the surface normal (obtained from the gradient of the SDF), $\gamma$ is the power decay, and $\mathbf{A}'_{\mathrm{tx}}$ is the transmitted input power. The proportional constant, $a$, in Eq. 2 remains fixed across all points on the ray. Therefore, without loss of generality, we absorb this constant into the transmitted amplitude term $A_{\mathrm{tx}}$, and define an effective transmitted amplitude $A'_{\mathrm{tx}}$.

**Opaque density representation** To represent volume density and accumulated transmittance, we start from the SDF. The scene is defined by an SDF $f : \mathbb{R}^3 \to \mathbb{R}$, which maps a spatial location $\mathbf{x} \in \mathbb{R}^3$ to its signed distance from the nearest surface of the object. This function is parameterized by a MLP. The object surface $S$ is defined as the zero-level set of the SDF.

Following [55], we use the S-density function $\phi_s(f(\mathbf{x}))$, defined as the derivative of the sigmoid function $\Phi_s(f(\mathbf{x}))$. The opaque density $\rho(u)$ is computed based on the rate of change of the occupancy field $\Phi_s$ along the ray direction.

**Signal representation** We employ a separate MLP that takes the 3D spatial position $\mathbf{x}$ as input to predict the reflection coefficient $\mathbf{a}$. The effective input amplitude $\mathbf{A}'_{\mathrm{tx}}$ is treated as a learnable global parameter, since the transmitted amplitude is fixed and consistent across all experiments.

**Discretization** We adopt the same discretization scheme as in [55], using alpha-blending for opaque density and transmittance estimation. The surface normal $\mathbf{n}$ is computed as the gradient of the SDF.

**Integration** According to Eq. 4, the signal received at each antenna is computed as an integration over the amplitude defined in Eq. 5. To evaluate this integral, we apply Monte Carlo sampling by tracing $N_{\mathrm{ray}}$ rays per antenna across space.

However, when combining all components, the overall computational complexity becomes prohibitively high. Assuming we sample $N_z$ depth points along each ray, and trace $N_{\mathrm{ray}}$ rays for each antenna. The MLP network incurs a computational cost of $C_{\mathrm{mlp}}$. The complexity of the signal tracing step for both forward and backpropagation is given by

$$\mathcal{O}(N_{\mathrm{ray}} N_z N_{\mathrm{ant}} C_{\mathrm{mlp}}). \tag{6}$$

In practice, to achieve 1mm resolution, the number of voxels becomes very large. If the synthetic aperture antenna array contains on the order of $10^5$ antennas, the total computation quickly reaches unrealistic limits. Even with GPU acceleration, a single forward pass in this setting has previously taken over one hour to compute. This scale far exceeds the number of available GPU threads and makes backpropagation-based learning infeasible under this brute-force computation strategy.

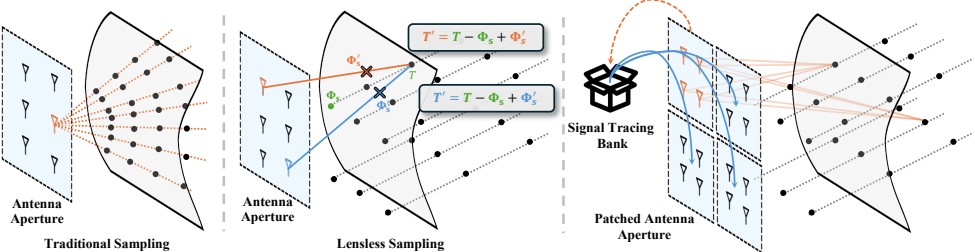

Figure 5: *Left:* Naïve sampling traces rays across the entire space for each antenna, resulting in the highest computational complexity. *Middle:* Lensless sampling strategy samples points along the radar's primary ray direction and reuses network predictions across antennas. Alpha blending and transmittance are calculated using lens-less alpha blending. *Right:* Signal Tracing Bank: subset of antennas is processed in each iteration, reducing computation for both forward and backward passes.

## 6 Lensless Sampling & Lensless Alpha Blending

To make volumetric rendering computationally feasible, we observe that the previous Monte Carlo sampling strategy contains significant redundancy. Specifically, rays emitted from different antennas frequently intersect at the same spatial locations, leading to duplicated computations. A straightforward solution might be to adopt uniform grid-based sampling in 3D space. However, this approach fails to preserve the structure of transmittance accumulation along individual rays, which is critical for accurate signal modeling.

To address both redundancy and physical correctness, we propose lensless sampling. The key insight is to share the computation of opaque density and accumulated transmittance across different antennas, where $\rho(u)$ and $T(u)$ only need to be computed once for each position, for all antenna rays that intersect the pose.

**Primary Ray** We first sample parallel rays aligned with the radar's primary direction $\omega_p$. The starting points $\mathbf{x}_{\mathrm{apr}}$ of these rays are sampled within the antenna aperture, and points are then sampled along each ray. Initially, we ignore the fact that $\rho(u)$ and $T(u)$ depend on the real ray direction, effectively replacing all individual receive directions $\omega_r$ with the shared primary direction $\omega_p$. The sampled points are formulated as $\mathbf{x} = \mathbf{x}_{\mathbf{apr}} + u\mathbf{p}$, making the assumption that these values are shared across all rays that pass through the same spatial location. Specifically, for each position, we compute $\rho(u)$ and $T(u)$ by using [55] once and reuse them for all antennas. We choose the radar's primary direction as the ray direction in order to minimize the discrepancy between the approximate and the actual rays emitted from the antennas.

**Lensless Alpha Blending** Neglecting the dependencies of $\rho(u)$ and $T(u)$ on the ray direction can, in principle, lead to inaccuracies. However, we prove in Appendix B.1 that when computing $\alpha_i = 1 - \exp\left(-\int_{u_i}^{u_{i+1}} \rho(u)\, \mathrm{d}u\right)$, the influence of the ray direction cancels out, and $\alpha_i$ remains identical to the value computed along the primary ray.

Furthermore, for the accumulated transmittance, we can show that, starting from the logistic distribution, the difference in $T(u)$ across different ray directions is equal to the difference in sigmoid-SDF values at the ray origins. Details of the derivation are provided in Appendix B.2. Formally, the transmittance along a real ray direction $T(u')$ can be efficiently derived from the transmittance along the primary ray direction $T(u)$ by adjusting with the sigmoid-SDF values at the respective ray origins, rather than computing a ray-dependent product of accumulated alpha values:

$$T(u') = T(u) - \Phi_s(f(\mathbf{x}(u_s))) + \Phi_s(f(\mathbf{x}(u'_s))), \tag{7}$$

where $\Phi_s(\cdot)$ denotes the cumulative density function (CDF) of logistic distribution, and $\mathbf{x}(u_s)$, $\mathbf{x}(u'_s)$ are the starting points of the respective rays. The power decay term $\gamma$, as well as the time and phase components, are computed deterministically using the actual rays. Additionally, the Sigmoid-SDF value at the ray origin, used to correct accumulated transmittance, is *excluded from backpropagation*, ensuring computational efficiency.

**Signal Tracing Bank** In Eq. E.2, we iterate over sampled points, and for each point, we perform signal tracing across all antennas. Within each signal tracing operation, we again loop over all sampled points along the ray. Although these computations are unavoidable, we can significantly

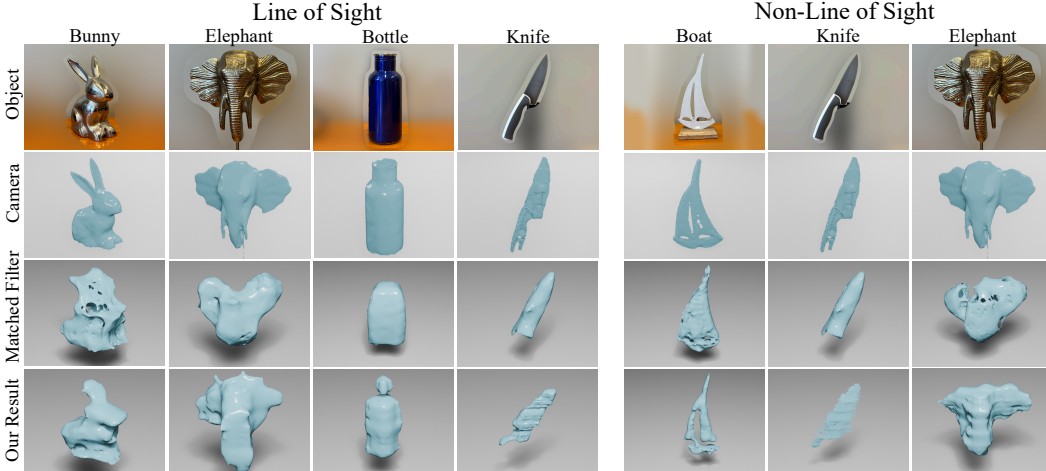

Figure 6: *Left*: Experimental results of objects with nothing between the radar and the object. *Right*: Results when the object is occluded from the radar. The camera scan done in line of sight.

reduce the computational burden by leveraging the fact that the matched filter contains no trainable parameters. This allows us to reuse signal tracing results of most antennas from previous iterations. During training, we implement a memory bank mechanism. We divide the antenna aperture into patches, analogous to convolutional processing. In each training iteration, signal tracing is performed only for antennas within a single patch, while the signal tracing results for antennas in the other patches are retrieved from the memory bank. After each iteration, the memory bank is updated with the newly computed results for the current patch.

**Computational complexity** By reusing the primary rays, we only need to evaluate the shared opaque density and accumulated transmittance $\mathcal{O}(N_{\text{ray}}N_{\text{z}}C_{\text{mlp}})$ times for both the forward and backward passes. The lensless approximation affects only the forward computation, and with the introduction of the Signal Tracing Bank, the effective antenna size becomes constant per iteration. Thus, the overall learnable component complexity is reduced to $\mathcal{O}(N_{\text{ray}}N_{\text{z}}C_{\text{mlp}})$. For the deterministic components of signal tracing and matched filtering, the computation time remains bounded by $\mathcal{O}(N_{\text{ray}}N_{\text{z}}N_{\text{t}})$ for both forward and backward propagation. This results in a total complexity that is significantly lower than the original formulation given in Eq. 6.

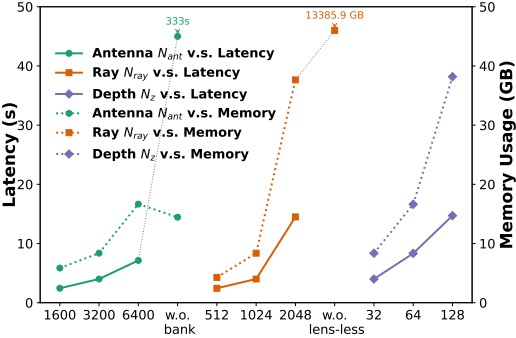

Figure 7: *Solid lines* indicate latency (time per iteration), while *dashed lines* show GPU memory usage. *Green* represents the configuration on patch antenna size in the Signal Tracing Bank. *Orange* represents the ray number, and "w.o. lens-less" refers to disabling lens-less sampling, whose memory usage is too large to measure experimentally and is instead deduced from theory. *Purple* represents the number of sample points on each ray.

Fig. 7 presents the latency and GPU memory usage across different sampling configurations. The baseline is chosen with $N_{\text{ray}} = 1024$, $N_z = 32$, and a patch size of 3200 antennas for the Signal Tracing Bank. Without lensless sampling, training becomes infeasible due to the excessive memory demand of simultaneously processing $N_{\text{ant}}N_{\text{ray}}N_z$ sample points. The Signal Tracing Bank drastically reduces both training time and memory consumption. We adopt this baseline configuration for all experiments to ensure efficiency.

# 7 Optimization

**Dynamic loss masking** Simply minimizing the difference between the rendered radar image and the ground truth using an L2 loss is not effective in radar imaging. This is because radar measurements are based on reflected signals, which are affected not only by the presence of volume density but also by the reflection direction. This means that a weak signal response in the rendered radar image can be caused by two distinct factors:

(1) low volume density at the sampled location, or (2) high volume density with the signal being reflected away from the receiver due to geometric misalignment. This inherent ambiguity makes direct supervision unreliable.

To address this, we introduce a masking strategy during training. If a sampled point has exhibited high matched filter response in any previous iteration but produces low power in the current iteration, we treat it as unreliable and exclude it from the loss computation. This prevents penalizing the model for physically valid signal paths that fail to reflect back to the receiver.

## 8 Experiments

**Dataset & Implementation** As there are no publicly available radar datasets for near range imaging, we collected our own dataset using a TI 1843BOOST mmWave radar [53] with a DCA1000EVM for raw data collection, attached to a Franka Research 3 controlled via the FrankaPy library [64]. We emulate 2D arrays by moving the radar with the robotic arm for 17-68 array viewpoints per object. For ground truth collection we used the Scaniverse App with an iPhone, which uses camera and LiDAR scans to compute 3D meshes and pointclouds. We trained our model for 50,000 iterations over 32 hours on a single NVIDIA H100 GPU. See the Appendix C for detailed data description and training hyperparameters.

Table 1: Quantitative results (*occluded).

| Object | F1↑ | | CD(mm)↓ | |
|---|---|---|---|---|
| | MF | Ours | MF | Ours |
| Wrench | 0.52 | **0.88** | 0.29 | **0.07** |
| Bunny | 0.56 | **0.81** | 0.03 | **0.01** |
| Elephant | 0.65 | **0.79** | 0.25 | **0.14** |
| Bottle | 0.44 | **0.97** | 0.35 | **0.05** |
| Knife | 0.44 | **0.71** | 0.59 | **0.49** |
| Star | 0.80 | **0.86** | 0.60 | **0.50** |
| Knife* | 0.45 | **0.83** | 0.84 | **0.26** |
| Elephant* | 0.42 | **0.52** | 0.52 | **0.46** |
| Boat* | 0.51 | **0.69** | 0.35 | **0.30** |
| Wrench* | 0.63 | **0.86** | 0.29 | **0.14** |

**Comparison to Baseline** As baseline, we compare results from GeRaF to the matched filter summation of all images. Since the corresponding output is a heatmap, we threshold the image and perform Poisson surface reconstruction on the point cloud. Qualitative comparisons are shown in Fig. 6. The object imaged is shown in the first row, followed by the camera scan, then the baseline matched filter reconstruction, and finally the output from GeRaF. The first four columns show reconstruction results when the object is in line of sight of the radar, and the last three columns show results for the object when it is occluded with a box or large sheet of paper. In the non-line-of-sight data processing, the box is cropped out of the MF image to just reconstruct inside the box. It is clear that the extracted meshes from the matched filter ground truth are not as complete as the outputs from GeRaF, and our method works even when vision methods would fail (eg. when line of sight is completely blocked).

Though the output of GeRaF is perhaps, not as detailed as the camera scans, this is primarily because our scans are limited to a single pitch axis, limiting the amount of reflections we can receive back from the objects, this limitation is discussed in more detail in Appendix F. Fig. 8 shows examples of the similarity between the rendered output and the ground truth radar heatmap. More qualitative results will be included in supplementary material.

Quantitative results are shown in Tab. 1, for F1-Score (F1) and the Chamfer Distance (CD) in millimeters, explained in Appendix C, where points were sampled uniformly on the camera mesh, the matched filter mesh and GeRaF's mesh. We can see that our method outperforms the Matched Filter (MF) baseline in both F-Score and Chamfer distance with a slight drop in accuracy when the items are moved into the box. More ablation studies are included in supplementary material.

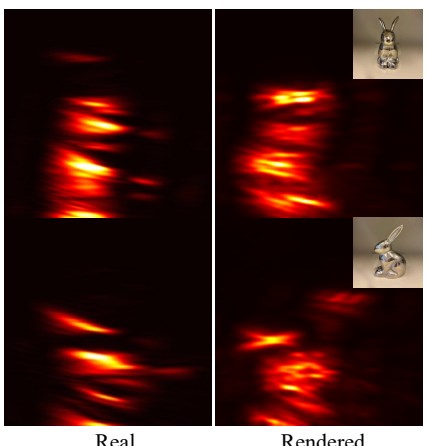

Real      Rendered

Figure 8: Rendered heatmaps compared to real heatmaps.

**Novel View Synthesis** We also demonstrate GeRaF's capability in novel view synthesis (NVS) [39]. To evaluate this, we split the available views into training and test sets. GeRaF is trained using only the training views and evaluated on the held-out test views. To assess the quality of the synthesized matched filter images, we compute the Peak Signal-to-Noise

Ratio (PSNR) between the rendered outputs and the ground truth matched filter images. PSNR is a widely used metric for evaluating image reconstruction fidelity, where higher values indicate closer similarity to the reference. The Peak Signal-to-Noise Ratio (PSNR) is computed as:

$$\text{PSNR} = 10 \cdot \log_{10} \left( \frac{MAX^2}{\text{MSE}} \right),\tag{8}$$

where $MAX$ is the maximum possible pixel value of the image, and MSE is the mean squared error between the synthesized image and the ground truth. Unlike RGB images, matched filter (MF) images do not have a fixed maximum pixel value. To make PSNR computation meaningful, we normalize each MF image by dividing it by its maximum value. As a result, the maximum possible pixel intensity is set to $MAX = 1.0$ for all PSNR evaluations.

We present four novel view synthesis results on the Bunny object in Fig. 9. The selected views include the front, left, back, and right sides of the object. In each case, GeRaF produces visualizations that closely match the ground truth. Quantitatively, the synthesized matched filter images achieve PSNR values around 30 dB, demonstrating the feasibility and effectiveness of GeRaF for novel view synthesis tasks.

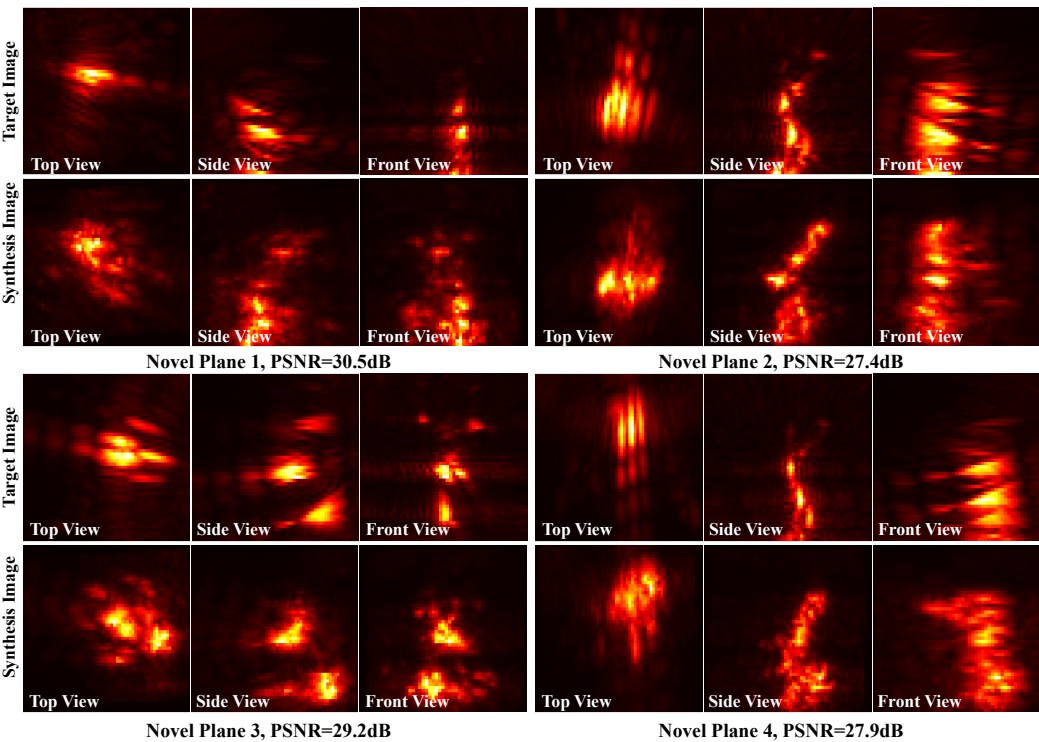

Figure 9: Novel view synthesis on four selected novel planes. We visualize the 3D matched filter images by projecting them along three axes using maximum intensity projection.

## 9  Conclusion

In this paper, we proposed the first method for applying differentiable rendering to radio frequency in the context of high resolution geometry reconstruction. Our method addresses the fundamental differences between RF and optical imaging, and we show evaluation on real captured data. We proposed a physics-aware volumetric rendering pipeline that incorporates radar-specific signal propagation and reflection models and a novel lensless sampling strategy to significantly reduce computational cost while maintaining resolution. Though there are limitations, discussed in Appendix F, we believe that this paper serves as the **first step** towards making millimeter-level geometry reconstruction feasible with neural representation learning.

## Acknowledgments

We thank the anonymous reviewers and members of the SENS Lab for their valuable feedback. We would also like to thank Ralf Boehnke, Dymtro Rachkov and Daniel Ardila Palomino from Sony Research for their helpful feedback. This project is funded in part by the Sony Faculty Innovation Fellowship.

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

# Appendix

This appendix is organized as follows:

- Section A provides more extensive radar background information.
- Section B provides the proof for our lenless alpha blending.
- Section C describes in detail the experimental parameters for collecting data.
- Section D describes the setup for the Novel View Synthesis experiment.
- Section E presents ablation studies.
- Section F goes into detail on limitations, future work and societal impact.

# A Radar Background

## A.1 Free-space Power Decay

In free space, the power of a radio frequency signal attenuates proportionally to the square of the distance it travels due to spherical spreading. When considering a round-trip path, where the signal travels from the transmitter to a point in space and then reflects back to the receiver, the decay is even more pronounced. This is known as *round-trip free-space path loss*, and the power decay factor reflected from distance $u$ is given by:

$$P_{\text{rx}} \propto \frac{1}{(4\pi u)^4} P_{\text{tx}} \tag{9}$$

This expression accounts for two instances of inverse-square spreading: one during transmission to the point and another during reflection back to the receiver. As such, the received power decreases proportionally to $1/u^4$.

## A.2 Radar Reflections

Building off of Sec. 3, we primarily expect specular reflections, albeit some flexibility in the signals which return to the receiver. Even though a perfectly specular signal will only return to the receiver if the reflecting surface is normal to the transmitted chirp, in actuality, there is a margin in which the signal is not perfectly specular, yet some signal still returns. Similar to commons ways of modeling diffused reflections, we follow a shifted Lambertian model [44, 43], sometimes also called the directive model. We represent the incident direction by the unit vector $\omega_i$, the surface normal by $\mathbf{n}$, and the ideal specular reflection direction by:

$$\omega_o = \omega_i - 2(\mathbf{n} \cdot \omega_i)\mathbf{n} \tag{10}$$

The ray which returns to the receiver antenna is denoted as $\omega_r$, and is calculated by taking the unit vector that goes from the surface point to the receiver location. The received power is modulated by the directional scaling factor:

$$(\omega_o \cdot \omega_r) \tag{11}$$

following the scaled Lambertian model mentioned above.

## A.3 Near Field vs. Far Field

Wireless imaging systems operate differently depending on the distance between the sensor and the target. This is denoted as near-field (or near range) and far-field (or far range) imaging. When an object is in the far-field, this is categorized as when the distance between the object and the radar are much larger than the wavelength and the aperture of the antenna system. For example, 10's of meters away. In this case,

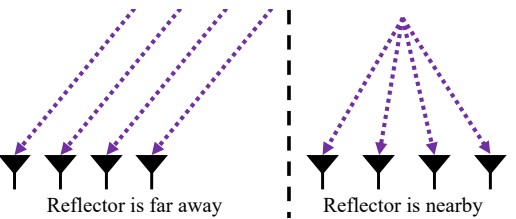

Figure 10: Difference between far-field and near-field rays. ***On the left,*** it's shown that the incoming rays can be approximated as nearly parallel, while ***on the right***, when the object is close to the antennas, we can no longer make this assumption.

the incoming waves can be *approximated* as planar [44]. Meaning all the incoming signals are parallel to each other. This significantly simplifies the image reconstruction, assuming the antennas are uniformly spaced, because (1) the corresponding filter weights can be reused across depth and (2) the weights are uniform, allowing us to use *beamforming* and Fourier Transforms to speed up computation time [3]. However, operating in the far-field comes at the cost of lower resolution reconstruction, since the RF waves spread out the further it travels.

On the other hand, in this paper, we are operating in the near-field. This mean the object is much closer to the antenna aperture, within a meter in our case. Importantly, this means that the incoming waves that are reflected from the object in the scene can no longer be approximated as parallel. The difference in signal paths is illustrated in Fig. 10. In this case, it is required that the exact signal propagation paths are taken into account, otherwise the radar images appear distorted [57, 45]. To correctly reconstruct an undistorted radar image, we *need* to use a matched filter (see Sec. 3), which comes with a high computational complexity, albeit a higher resolution image. For example, in Fig. 11 a heatmap of a wrench is shown when processed with beamforming vs. using the matched filter.

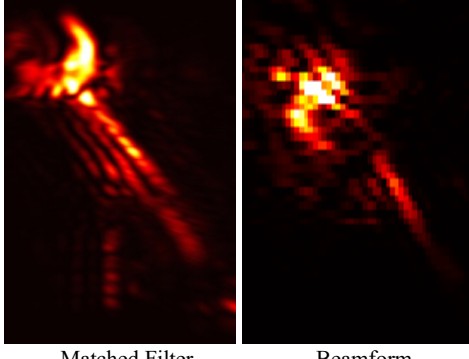

Matched Filter          Beamform

Figure 11: Difference between processing radar data captured in near-field with matched filter vs beamforming.

## A.4 Differentiable Matched Filter & Signal Tracing

We have the forward path of the Matched Filter shown in Eq. E.2:

$$P(\mathbf{x}_j) = \left\| \sum_{i=1}^{N_{\text{ant}}} \sum_{t} s(i,t) \cdot e^{j2\pi k \tau_i t} \cdot e^{j2\pi f \tau_i} \right\|, \mathbf{x}_j \in \Omega_{\text{pts}},$$

The backpropagated gradient to the signal $s(i,t)$ is given by:

$$\frac{\partial L}{\partial s(i,t)} = \sum_{\mathbf{x}_j \in \Omega_{\text{pts}}} \frac{1}{P(\mathbf{x})} \cdot \frac{\partial L}{\partial P(\mathbf{x})} \cdot e^{-j2\pi k \tau_i t} \cdot e^{-j2\pi f \tau_i} \tag{12}$$

Signal Tracing is given by Eq 4,

$$s(i,t) = \sum_{\mathbf{x}_j \in \Omega_{\text{pts}}} A_{\text{rx}}(\mathbf{x}_j) \, e^{-j2\pi k \tau_j t} \, e^{-j2\pi f \tau_j},$$

The backpropagated gradient to the amplitude $A_{\text{rx}}(\mathbf{x}_j)$ is given by:

$$\frac{\partial L}{\partial A_{\text{rx}}(\mathbf{x}_j)} = \sum_{i=1}^{N_{\text{ant}}} \sum_{t} \frac{\partial L}{\partial s(i,t)} \cdot e^{j2\pi k \tau_i t} \cdot e^{j2\pi f \tau_i} \tag{13}$$

## A.5 Code Implementation

**Differentiable Matched Filter** We implement both the forward and backpropagation of the matched filter in CUDA using parallel computation. The corresponding pseudocode is shown in Alg. 1.

---

**Algorithm 1** Differentiable Matched Filter Algorithm

---

**[1] Forward:** Input signal $s(i,t)$ where $i = 0, \ldots, N_{\text{ant}} - 1, t = 0, \ldots, N_t - 1$;
sampling points $\mathbf{x}_j$ where $j = 0, \ldots, N_{\text{ray}}N_{\text{s}} - 1$
**for Parallel** $j = 0, \ldots, N_{\text{ray}}N_{\text{s}} - 1$ **do**
    $P(\mathbf{x}_j) \leftarrow 0$
    **for** $i = 0, \ldots, N_{\text{ant}} - 1$ **do**
        **for** $t = 0, \ldots, N_t - 1$ **do**
            $P(\mathbf{x}_j) \leftarrow P(\mathbf{x}_j) + s(i,t) \cdot e^{j2\pi k\tau_i t} \cdot e^{j2\pi f\tau_i}$
    $P(\mathbf{x}_j) \leftarrow \|P(\mathbf{x}_j)\|$
**Output:** $P(\mathbf{x}_j)$ for all $j$

**[2] Backward:** Gradient of matched filter power $\frac{\partial L}{\partial P(\mathbf{x}_j)}$ for $j = 0, \ldots, N_{\text{ray}}N_{\text{s}} - 1$;
Input signal $s(i,t)$ where $i = 0, \ldots, N_{\text{ant}} - 1, t = 0, \ldots, N_t - 1$;
Sampling points $\mathbf{x}_j$ where $j = 0, \ldots, N_{\text{ray}}N_{\text{s}} - 1$
**for Parallel** $i = 0, \ldots, N_{\text{ant}} - 1$ **do**
    **for Parallel** $t = 0, \ldots, N_t - 1$ **do**
        $\frac{\partial L}{\partial s(i,t)} \leftarrow 0$
        **for** $j = 0, \ldots, N_{\text{ray}}N_{\text{s}} - 1$ **do**
            $\frac{\partial L}{\partial s(i,t)} \leftarrow \frac{\partial L}{\partial s(i,t)} + \frac{1}{P(\mathbf{x}_j)} \cdot \frac{\partial L}{\partial P(\mathbf{x}_j)} \cdot e^{-j2\pi k\tau_i t} \cdot e^{-j2\pi f\tau_i}$
**Output:** $\frac{\partial L}{\partial s(i,t)}$ for all $i, t$

---

**Signal Tracing** We implement both the forward and backpropagation in CUDA using parallel computation. The corresponding pseudocode is shown in Alg. 2.

---

**Algorithm 2** Signal Tracing

---

**[1] Forward:** Sampling points $\mathbf{x}_j$ where $j = 0, \ldots, N_{\text{ray}}N_{\text{s}} - 1$;
Amplitude $A_{\text{rx}}(\mathbf{x}_j)$ where $j = 0, \ldots, N_{\text{ray}}N_{\text{s}} - 1$
**for Parallel** $i = 0, \ldots, N_{\text{ant}} - 1$ **do**
    **for Parallel** $t = 0, \ldots, N_t - 1$ **do**
        $s(i,t) \leftarrow 0$
        **for** $j = 0, \ldots, N_{\text{ray}}N_{\text{s}} - 1$ **do**
            $s(i,t) \leftarrow s(i,t) + A_{\text{rx}}(\mathbf{x}) \cdot e^{-j2\pi k\tau_i t} \cdot e^{-j2\pi f\tau_i}$
**Output:** $s(i,t)$ for all $i, t$
**[2] Backward:** Gradient of signal $\frac{\partial L}{\partial s(i,t)}$ for $i = 0, \ldots, N_{\text{ant}} - 1, t = 0, \ldots, N_t - 1$
Sampling points $\mathbf{x}_j$ where $j = 0, \ldots, N_{\text{ray}}N_{\text{s}} - 1$
**for Parallel** $j = 0, \ldots, N_{\text{ray}}N_{\text{s}} - 1$ **do**
    $\frac{\partial L}{\partial A_{\text{rx}}(\mathbf{x}_j)} \leftarrow 0$
    **for** $i = 0, \ldots, N_{\text{ant}} - 1$ **do**
        **for** $t = 0, \ldots, N_t - 1$ **do**
            $\frac{\partial L}{\partial A_{\text{rx}}(\mathbf{x}_j)} \leftarrow \frac{\partial L}{\partial A_{\text{rx}}(\mathbf{x}_j)} + \frac{\partial L}{\partial s(i,t)} \cdot e^{j2\pi k\tau_i t} \cdot e^{j2\pi f\tau_i}$
**Output:** $\frac{\partial L}{\partial A_{\text{rx}}(\mathbf{x}_j)}$ for all $j$

---

## B  Lensless Alpha Blending

### B.1  Alpha

Following [55], the opaque density computed along the primary ray $\mathbf{p}$ is defined as:

$$\rho(u) = \max\left(\frac{-\frac{\mathrm{d}\Phi_s}{\mathrm{d}u}(f(\mathbf{x}(u)))}{\Phi_s(f(\mathbf{x}(u)))}, 0\right), \tag{14}$$

where $\phi_s(x)$ and $\Phi_s(x)$ denote the probability density function (PDF) and cumulative distribution function (CDF) of the logistic distribution, respectively.

The corresponding alpha value over the interval $[u_i, u_{i+1}]$ is then:

$$\alpha_i = 1 - \exp\left(-\int_{u_i}^{u_{i+1}} \rho(u)\mathrm{d}u\right) = 1 - \exp\left(-\int_{u_i}^{u_{i+1}} \frac{-\frac{\mathrm{d}\Phi_s}{\mathrm{d}u}(f(\mathbf{x}(u)))}{\Phi_s(f(\mathbf{x}(u)))}\,\mathrm{d}u\right). \tag{15}$$

When changing from the primary ray $\mathbf{p}$ to a real ray $\mathbf{r}$, the derivative term in the numerator transforms as:

$$-\frac{\mathrm{d}\Phi_s}{\mathrm{d}u}(f(\mathbf{x}(u))) = -\frac{\mathrm{d}\Phi_s}{\mathrm{d}u'}(f(\mathbf{x}(u))) \cdot \left|\frac{\mathrm{d}u'}{\mathrm{d}u}\right| = \rho'(u') \cdot \left|\frac{\mathrm{d}u'}{\mathrm{d}u}\right|, \tag{16}$$

where $u$ and $u'$ are the sampling parameters along the primary and real rays, respectively, and $\rho'(u')$ denotes the opaque density computed along the real ray.

In integration, we have:

$$\alpha_i = 1 - \exp\left(-\int_{u_i}^{u_{i+1}} \rho(u)\,\mathrm{d}u\right) = 1 - \exp\left(-\int_{u_i}^{u_{i+1}} \rho(u)\,\mathrm{d}u' \cdot \left|\frac{\mathrm{d}u}{\mathrm{d}u'}\right|\right). \tag{17}$$

By substituting the transformed opaque density from Eq. 16 into Eq. 17, we obtain:

$$\alpha_i = 1 - \exp\left(-\int_{u'_i}^{u'_{i+1}} \rho'(u')\,\mathrm{d}u'\right) = \alpha'_i, \tag{18}$$

demonstrating that the computed $\alpha_i$ remains unchanged when transitioning from the primary ray to the real ray.

## B.2 Transmittance

Following [55], in the primary ray direction $\mathbf{p}$, we define the transmittance as:

$$T(u)\rho(u) = |\nabla f(\mathbf{x}(u)) \cdot \mathbf{p}|\, \phi_s(f(\mathbf{x}(u))) = -\frac{\mathrm{d}\Phi_s}{\mathrm{d}u}(f(\mathbf{x}(u))), \tag{19}$$

where $\phi_s(\cdot)$ and $\Phi_s(\cdot)$ are the PDF and CDF of the logistic distribution, respectively.

Given the transmittance is defined as $T(u) = \exp\left(-\int_0^u \rho(v)\,\mathrm{d}v\right)$, differentiating both sides yields:

$$\frac{\mathrm{d}T}{\mathrm{d}u} = \frac{\mathrm{d}}{\mathrm{d}u}\exp\left(-\int_0^u \rho(v)\,\mathrm{d}v\right) = -\rho(u)\exp\left(-\int_0^u \rho(v)\,\mathrm{d}v\right) = -\rho(u)T(u),$$

which implies:

$$T(u)\rho(u) = -\frac{\mathrm{d}T}{\mathrm{d}u}(u). \tag{20}$$

Combining Eq. 19 and Eq. 20, we obtain:

$$-\frac{\mathrm{d}\Phi_s}{\mathrm{d}u}(f(\mathbf{x}(u))) = -\frac{\mathrm{d}T}{\mathrm{d}u}(u). \tag{21}$$

Because the geometry-dependent signed distance function (SDF) does not vary with ray direction, we assume the boundary condition:

$$\Phi_s(f(\mathbf{x}(u))) = \Phi_s(f(\mathbf{x}(u'))).$$

Then, given the starting CDF values $\Phi_s(f(\mathbf{x}(u_s)))$ on the primary ray $\mathbf{p}$ and $\Phi_s(f(\mathbf{x}(u'_s)))$ on the real ray $\mathbf{r}$, integrating Eq. 21 leads to:

$$T(u') - T(u) = \Phi_s(f(\mathbf{x}(u'_s))) - \Phi_s(f(\mathbf{x}(u_s))), \tag{22}$$

which completes the proof of Eq. 7.

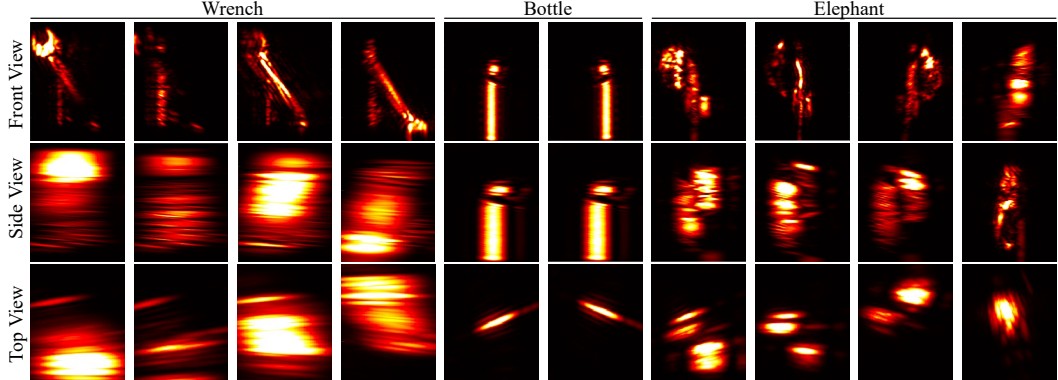

Figure 12: Subset of matched filter images used as ground truth in GeRaF. The first row shows the radar heatmaps summed across the depth, giving a front view of the object relative to GeRaF's coordinate system, the second row is summed across the horizontal axis (azimuth), giving a side view and the last row is summed along the vertical axis (elevation), giving a top down or bird's eye view.

## C   Experiment Details

**Data Collection Setup** We collected our own dataset using a TI 1843BOOST mmWave radar [53] with a DCA1000EVM for raw data collection, attached to a Franka Research 3 controlled via the FrankaPy library [64]. We emulate 2D arrays by moving the radar with the robotic arm for 17-68 (more scans were made for more complex objects) different array viewpoints, which are each 0.14m $\times$ 0.25m in size with an antenna spacing of $\sim (\frac{\lambda}{4} - \frac{\lambda}{2})$ between all antennas. Each imaging plan was shifted circularly (yaw) around the center of the objects location to measure various viewpoints. A subset of the ground truth heatmaps are visualized in Fig. 12. Though the actual groundtruth used in GeRaF is a 3D heatmap, for ease of visualization, we plot the 3D heatmap summed along each of the 3 dimensions (depth, azimuth, elevation), respectively. Each of the columns show the radar heatmap from a different scanning plane.

In order to capture $360°$ scans, we use a rotation platform to rotate the object which was placed aroun 0.3-0.5m away. The radar parameters were set to have a starting frequency of 77 GHz usable sweeping bandwidth of 3.59 GHz, giving $0.04m$ in range resolution. The slope used is $70.15(MHz/\mu s)$ capturing 64 ADC samples with a 1250ksps sampling rate for each chirp.

**Training Setup** For the SDF Network, we use an MLP with 8 layers and a hidden dimension of 256. We apply sinusoidal positional encoding with 10 frequency levels as input. The Reflectivity Network is implemented as an MLP with 4 layers and a hidden dimension of 256. The signal power prediction is implemented as a single trainable parameter.

We trained our model for 50,000 iterations over 32 hours on a single NVIDIA H100 GPU, using `mmDetection3D` [10] as the code base. We used an initial learning rate of $1 \times 10^{-3}$, but due to the sparsity of the input, the learning rate for the SDF Network was reduced to $1 \times 10^{-4}$. Training was performed using the AdamW optimizer with cosine annealing learning rate scheduling.

**Metric Calculation** The metrics used to compare GeRaF and the matched filter baseline against the camera ground truth are the F1-Score (evaluated with $\tau = 0.01$) and Chamfer distance. These two metrics evaluate the similarity between two 3D point clouds. For mesh to point cloud conversion, we sample uniformly along the meshes for each of the three candidates with 5000 points. We then align the matched filter point clouds to the camera baseline and align GeRaF's point clouds to the camera baseline.

The F1-Score is calculated by finding the nearest point in the camera point cloud and computing the distance in both directions. Precision and recall are computed as the proportions of these distances that fall below $\tau$, and the F1-Score is derived by:

$$F_1 = 2 \cdot \frac{\text{Precision} \cdot \text{Recall}}{\text{Precision} + \text{Recall}} \tag{23}$$

The Chamfer distance computes the pairwise distance between two pointclouds by finding the nearest neighbors for each point between point clouds and calculating the squared distance, returning the average distance. This is calculated in both directions.

## D  Novel View Synthesis

We also demonstrate GeRaF's capability in novel view synthesis (NVS) [39]. To evaluate this, we split the available views into training and test sets. Figure 13 illustrates the detailed data split. The results are shown in Fig. 9.

## E  Ablation Studies

### E.1  Number of Radar Scanning Planes

We study the effect of varying the number of radar scanning planes (analogous to the number of views in vision-based reconstruction) used during training. A total of 104 planes were measured to cover a full 360-degree view of the object. We uniformly sampled subsets of these planes to simulate different input settings.

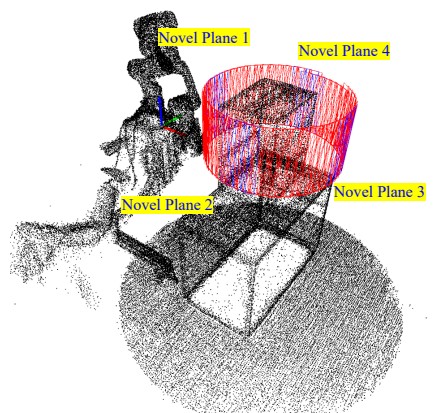

Figure 13: Novel view synthesis setting. The blue planes represent novel views that are not included during training.

Even with only 8 planes, GeRaF is able to reconstruct the basic shape of the Bunny, though the result contains noticeable noise and inaccurate details. When the number of planes is fewer than 40, the reconstructed shape becomes visibly distorted, and the noise level increases significantly. With more than 72 planes, the overall shape is correctly reconstructed, and differences are mostly limited to fine details. This indicates that GeRaF is robust to the number of input planes once a sufficient coverage threshold is met.

Overall, the results show a clear trend: increasing the number of radar scanning planes improves the reconstruction quality, yielding more accurate geometry and less noise.

### E.2  Number of Temporal Sample

We study the effect of the number of temporal samples $N_t$ on geometry reconstruction quality. According to the matched filter equation:

$$P(\mathbf{x}) = \left\| \sum_{i=1}^{N_{\text{ant}}} \sum_{t=1}^{N_t} s(i,t)\, e^{j2\pi k \tau_i t}\, e^{j2\pi f \tau_i} \right\|,$$

increasing the number of temporal samples improves resolution. However, the computational cost of the matched filter increases with $N_t$.

To assess this trade-off, we evaluate reconstruction quality under different values of $N_t$, as shown in Fig. 15. Empirically, we observe no significant degradation in performance as $N_t$ varies, demonstrating GeRaF's robustness to noise. While using fewer temporal samples can theoretically reduce the quality of the ground

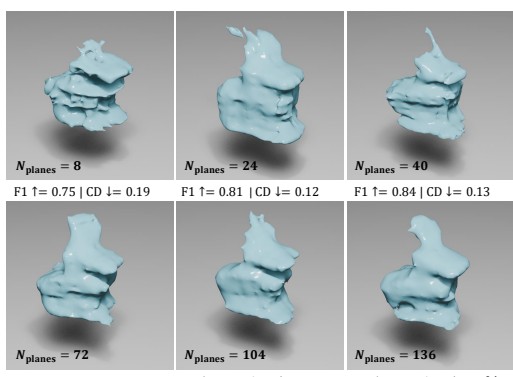

Figure 14: Ablation study on the number of radar scanning planes. F1 denotes the F1-Score and CD denotes the Chamfer Distance. ↑ indicates higher values are better, while ↓ indicates lower values are better.

truth matched filter image, the increase in latency with larger $N_t$ remains relatively limited. Considering the balance between performance and efficiency, we set $N_t = 64$ in our final implementation.

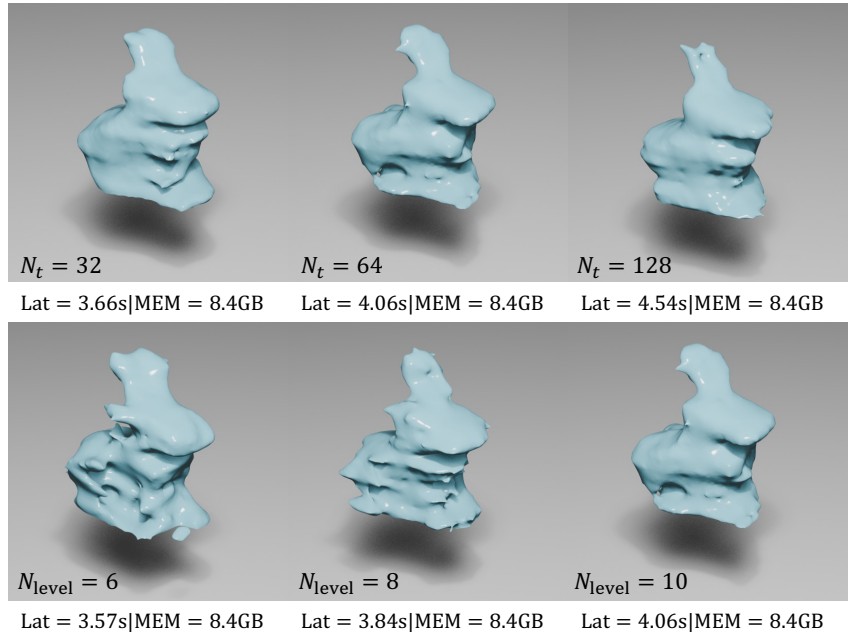

$N_t = 32$     $N_t = 64$     $N_t = 128$

Lat = 3.66s|MEM = 8.4GB  Lat = 4.06s|MEM = 8.4GB  Lat = 4.54s|MEM = 8.4GB

$N_{\text{level}} = 6$    $N_{\text{level}} = 8$    $N_{\text{level}} = 10$

Lat = 3.57s|MEM = 8.4GB  Lat = 3.84s|MEM = 8.4GB  Lat = 4.06s|MEM = 8.4GB

Figure 15: ***Top:*** Ablation study on the number of temporal samples $N_t$. ***Bottom:*** Ablation study on the positional encoding frequency level $N_{\text{level}}$. *Lat* denotes latency, and *MEM* denotes GPU memory usage.

### E.3 Positional Encoding Frequency Level

We apply sinusoidal positional encoding as in NeRF [39], where each input $x$ is mapped to a higher-dimensional space using:

$$\gamma(x) = \left[\sin(2^0\pi x), \cos(2^0\pi x), \ldots, \sin(2^{N_{\text{level}}-1}\pi x), \cos(2^{N_{\text{level}}-1}\pi x)\right],$$

with $N_{\text{level}}$ controlling the number of frequency bands.

Low frequency levels (e.g., $2^0, 2^1$) capture coarse, smooth variations, while high frequency levels (e.g., $2^9, 2^{10}$) capture fine, high-frequency details such as sharp edges or thin structures. The number of frequency levels $N_{\text{level}}$ controls the trade-off between expressiveness and overfitting: a higher $N_{\text{level}}$ enables modeling of finer details, whereas a lower $N_{\text{level}}$ reduces model complexity and helps prevent overfitting to noise.

To assess this trade-off, we evaluate reconstruction quality under different positional encoding frequency levels $N_{\text{level}}$, as shown in the bottom of Fig. 15.

We observe that higher frequency levels enable better reconstruction of fine details in the Bunny and improve robustness to noise in radio frequency signals. Additionally, increasing $N_{\text{level}}$ leads to higher computational latency, but the increase is relatively limited. Considering this trade-off, we set $N_{\text{level}} = 10$ in our final configuration.

### E.4 Resolution

In Section 6 of the main submission, we discuss the computational resources required by different training-time sampling resolutions.

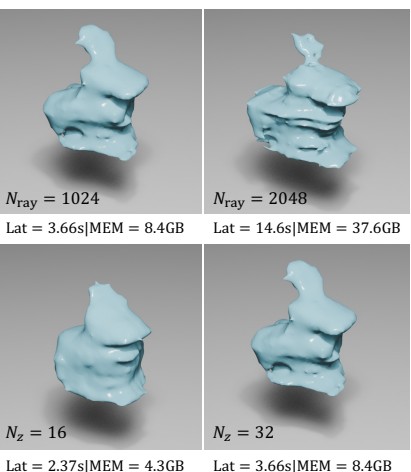

$N_{\text{ray}} = 1024$  $N_{\text{ray}} = 2048$

Lat = 3.66s|MEM = 8.4GB Lat = 14.6s|MEM = 37.6GB

$N_z = 16$   $N_z = 32$

Lat = 2.37s|MEM = 4.3GB Lat = 3.66s|MEM = 8.4GB

Figure 16: ***Top:*** Ablation study on the number of ray samples $N_{\text{ray}}$. ***Bottom:*** Ablation study on the number of depth samples $N_z$. *Lat* denotes latency, and *MEM* denotes GPU memory usage.

For the number of sampling rays, results show that increasing the number of rays leads to higher memory usage and longer latency, but surprisingly worse reconstruction performance. A possible explanation is that in radio frequency data, denser sampling introduces more noise, which slows convergence. As a result, under a fixed number of training iterations, higher sampling resolutions may lead to suboptimal solutions due to insufficient convergence. Based on these observations, we set $N_{\text{ray}} = 1024$ in our final configuration.

For the number of depth samples $N_z$, results indicate that fewer depth samples lead to lower reconstruction detail and a coarser overall shape. Although reducing depth samples improves training efficiency linearly in time, it comes at the cost of reduced fidelity. Therefore, we set $N_z = 32$ as our final configuration to balance quality and performance.

### E.5 Dynamic loss masking

In Section 7 of the main submission, we discuss the motivation for using dynamic loss masking. The primary reason is the inherent ambiguity in interpreting low-power regions in matched filter (MF) imaging. Specifically, a weak signal response in the rendered radar image may arise from two distinct causes: (1) low volume density at the sampled location, or (2) high volume density with the signal being reflected away from the receiver due to geometric misalignment. This ambiguity makes direct supervision unreliable in low-reflection regions.

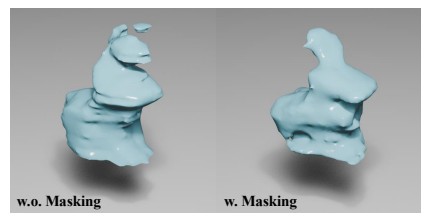

Figure 17: Ablation study on dynamic loss masking. "w.o." denotes without masking, and "w." denotes with masking.

As shown in the top of Fig. 17, training without dynamic loss masking leads to incorrect reconstruction. The model tends to interpret low-intensity regions as low density, resulting in artifacts—such as a distorted ear in the Bunny example—where the true geometry reflects weakly due to its orientation rather than material absence. This highlights the importance of masking unreliable regions during training to improve geometric fidelity.

### E.6 Effect of Radar physics

We introduce radar physics in Section 3 of the main submission. Radar reflections are dominated by specular effects and are commonly approximated using the Lambertian model [44]. In this section, we study the effect of incorporating the Lambertian model into our formulation.

To isolate its impact, we remove the $\omega_o \cdot \omega_r$ term from Eq. 5 in the main submission. As shown at the Fig. 18, the visual difference between using and not using the Lambertian model appears minor. However, quantitative results indicate that omitting the Lambertian term leads to worse performance. Specifically, in the left-side view, the reconstructed bottom of the Bunny appears significantly larger than in the right-side view, contributing to higher Chamfer Distance error.

That said, the Lambertian model does not fully capture the physical behavior of radar reflections, especially under specular and material-dependent conditions. Thus, while it provides a useful approximation, the Lambertian model is not an optimal representation of true radar physics.

## F   Discussion

### F.1   Potential Societal Impacts

There are a few potential negative impacts that RF neural representations could cause. Being able to perform 3D reconstruction of things behind occlusions could potentially raise concerns for privacy. Additionally like many other learning based work, the heavy compute power raises concerns over the environment sustainability. However, this work could also potentially help with things like suspicious package detection.

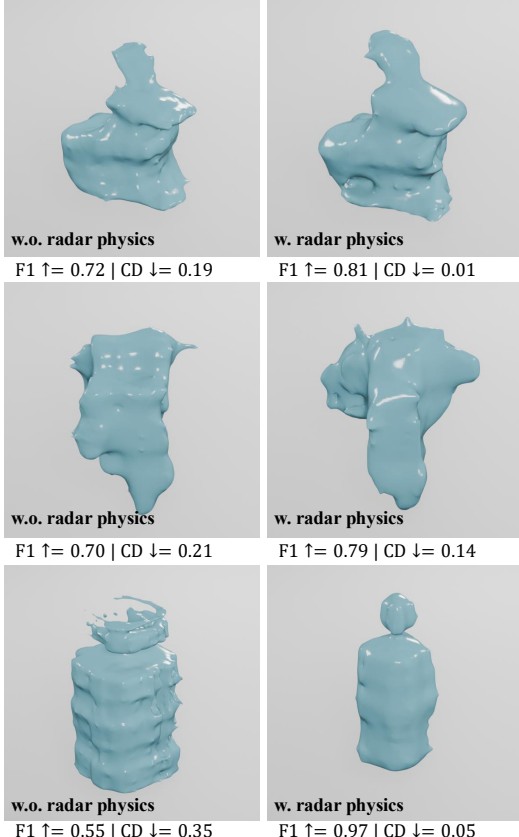

Figure 18: Ablation study on radar physics. "Without radar physics" indicates that the Lambertian model [44] is not used.

## F.2   Limitations & Future Work

**Radar Scans** Our method currently is limited by the diversity of the networks radar heatmap ground truths collected. More specifically, if there was a point that existed on the surface of the object, which never reflects back to the radar in any of the scans taken, then the network has no way of creating a point in that location, because it has never received any reflection from that location, making it invisible. Future work is required for dealing with unseen surfaces, and more comprehensive scanning methods.

**Materials** We currently tested the system with objects made of metals, which allows for more clear reflections. On the other hand, metal objects are *very* specular, meaning more radar scans are required to receive signals from all parts of the object. Dealing with objects made of other materials or a composition of materials is left for future work, which will require significantly adjusting the radar reflection model.

**Thick Occlusions** We have shown results for some occlusions (cardboard, paper), however, in the case of thicker occlusions this problem is not so straightforward since this requires the network to predict *two or more* surfaces rather than ignoring the weak occlusion reflections. Future work is required to extend the surface reconstruction to deal with multiple surfaces.

**Better Radio Frequency Model** As discussed in the radar physics section, unlike vision-based inverse rendering which often adopts the Cook-Torrance BRDF [11], with components such as the Fresnel term, Schlick-GGX geometry function, and Trowbridge-Reitz GGX distribution to model diffraction and metallic reflections, the radio frequency (RF) domain still relies on the simplified Lambertian model. This model is not even sufficient for accurate forward rendering, leading to significant reflection errors. As a result, inverse rendering based on such a coarse approximation

cannot reliably reconstruct detailed geometry, since the underlying reflection model introduces substantial errors.

**Lower Computational Complexity** Even with the introduction of lens-less sampling and alpha blending, which make real-time differentiable rendering feasible, the computational cost remains significantly higher compared to vision-based rendering. In the vision domain, reflective models benefit from techniques such as split-sum approximation [41] and spherical harmonics [54], which avoid costly Monte Carlo sampling by operating efficiently over 1D rays.

However, directly applying these techniques to radar reflection is not feasible, since traditional alpha blending operates along 1D rays, whereas radar rendering requires sampling over a 3D space. This dimensional difference introduces substantial complexity.Therefore, future work should explore how to achieve 3D space sampling with computational complexity comparable to that of 1D ray-based sampling, in order to further improve the efficiency of RF-based differentiable rendering.

**Accurate multi-level surface representation** In vision-based geometry reconstruction, multi-level surfaces are typically not a concern, as surfaces occluded by others are not visible in RGB images and therefore do not contribute to reconstruction. However, this is not the case in radar sensing, where signals can penetrate and reflect off multiple layers of geometry, resulting in observable multi-level surfaces. As a result, representing and reconstructing such multi-layered structures becomes a key challenge in RF-based 3D reconstruction. Developing an effective multi-level surface representation is thus essential for fully leveraging the unique capabilities of radar.

