# OpenReview forum: "GeRaF: Neural Geometry Reconstruction from Radio Frequency Signals"
_NeurIPS.cc/2025/Conference — NeurIPS 2025 spotlight_

### Official Review · Reviewer_t7Hp · 2025-06-30

**Clarity:** 4
**Significance:** 3
**Originality:** 4
**Rating:** 4
**Confidence:** 3

**Summary:**

This paper's goal is near-range 3D geometry reconstruction from radio frequency signals. The authors challenge low SNR, and high computational frequency due to lensless imaging and propose novel RF volume rendering equations for different ray propagation physics of radio waves unlike visible light.
GeRaF framework consists of an SDF network and a reflective network. Each predicts geometry (volume density and normals) and reflectiveness. The authors assume every 3D point in space is a potential reflector, formulating the received signal as a sum of phase-shifted signals over sampled points(signal tracing). The amplitude of the signal is determined by the reflection direction computed from normal, the reflectivity of the 3D point, and squared transmittance due to the round-trip path together with effective input amplitude which is learned as a global parameter.
To render a lensless image in reduced complexity, the authors devise the lensless alpha blending strategy and signal tracing bank, This strategy utilizes primary ray direction to sample points and blending. Authors show that ray-dependency in transmittance and alpha computation can be safely removed and the use of primary ray direction can reduce redundant computation. The authors also adopt a memory bank mechanism that reuses the signal tracing results of previous iterations. Lastly, the authors proposed dynamic loss masking for loss backpropagation on valid signals.
Compared to the Match Filter baseline, GeRaF shows higher geometry reconstruction accuracy in both line-of-sight and non-line-of-sight.

**Questions:**

1. The main contribution of this work would be the validity of radar physics formulation. To see the validity of the model, can authors also include the ablation in the other scenes as well? (Section 2.6 of supplementary material)

**Ethical Concerns:**

["NO or VERY MINOR ethics concerns only"]

**Final Justification:**

Authors has well addressed my concerns and I was convinced that the novelty proposed by this paper outweighs weakness. Therefore I am retaining my score as positive to borderline accept.

**Limitations:**

Yes

**Quality:**

3

**Strengths And Weaknesses:**

Strength
1. The authors are the first to suggest a 3D reconstruction of an object in a near-range with radio frequency signals. This enables the reconstruction of an object in non-line-of-sight, which is a promising application.
2. This paper formulated a lensless neural rendering framework for RF signal images. The authors devised a strategy to compute computationally complex forward and backward operations of lensless imaging by removing the necessity of ray-direction dependency in the computation pipeline. The authors clearly derive their novel imaging formulation in detail
3. The computational complexity analysis graph shows that the latency and memory capacity are greatly reduced by author’s proposals.

Weakness.
1. Due to the absence of an existing public dataset on near-range radio frequency scanning and precedent works, the evaluation of the method is limited to the dataset captured by authors, and comparisons are conducted only with the Matched Filter baseline.

---

> ### Author Rebuttal · Authors · 2025-07-30
>
> We would like to thank the reviewer for their thoughtful comments. We especially welcome the reviewer's appreciation of both the promising applications of our work but also the technical novelty involved in getting neural implicit representations to work with RF rendering, specifically lensless imaging and lowering the computational complexities to make training feasible. In the following paragraphs, we respond to the comments and concerns raised by the reviewer.
>
> __The reviewer highlights an important issue that the evaluation is only on our dataset.__
>
> Unfortunately, there are no publicly available radar datasets that scan objects from multiple viewpoints. Additionally, we would like to add that the Matched Filter baseline is the algorithm which maximizes the signal-to-noise ratio in the recovered image [3], which produces the highest quality image given there is no prior information of the scene. Further, there are no works, to our knowledge, whose methods are directly applicable to this problem. However, we believe that the comparison to Matched Filter is sufficient given that this is the current state of the art in radar imaging. Nonetheless, we plan to explore a comparison of all of our results with the Range Migration Algorithm [5], Beamforming [4] (two slightly faster, but less accurate radar based reconstructions) and NeRO [2] (reflective material vision reconstruction) for another radar signal processing and vision based comparisons.
>
> __The reviewer asks for further evaluation showing the effect of radar physics.__
>
> The reviewer is correct, the effect of radar physics is a very important contribution to our work. We will include qualitative results on different scenes in the revised paper. For preliminary results we see that the elephant's Chamfer Distance increases from 0.14cm to 0.3cm and F1-Score decreases from 0.79 to 0.71 when radar physics are not taken into account. However, even more drastically, in the qualitative comparison the ears of the elephant completely disappear from the surface. We will include more results in the revised paper.
>
> We would like to once again thank the reviewer for the time and comments. We sincerely hope that their concerns and questions are clarified.
>
> [2] Liu, Yuan, et al. "Nero: Neural geometry and brdf reconstruction of reflective objects from multiview images." ACM Transactions on Graphics (ToG) 42.4 (2023): 1-22.
>
> [3] Novak, Leslie M., Gregory J. Owirka, and Christine M. Netishen. "Radar target identification using spatial matched filters." Pattern Recognition 27.4 (1994): 607-617.
>
> [4] Harter, Marlene, Andreas Ziroff, and Thomas Zwick. "Three-dimensional radar imaging by digital beamforming." 2011 8th European Radar Conference. IEEE, 2011.
>
> [5] Zhu, Rongqiang, et al. "Range migration algorithm for near-field MIMO-SAR imaging." IEEE Geoscience and Remote Sensing Letters 14.12 (2017): 2280-2284

---

> > ### Comment · Reviewer_t7Hp · 2025-08-06
> >
> > I have read the other reviewers' comments and the authors' rebuttal. I acknowledge the limited availability of public datasets and directly comparable methods. I appreciate the work's contributions in advancing a pioneering area. Therefore, I will keep my rating as is.

---

> > > ### Author Response · Authors · 2025-08-06
> > >
> > > Thank you once again to the reviewer for their thorough review and feedback, their comments are very important to us for improving our paper!

---

### Official Review · Reviewer_U8dg · 2025-07-02

**Clarity:** 2
**Significance:** 3
**Originality:** 3
**Rating:** 4
**Confidence:** 2

**Summary:**

This work proposes a novel differentiable rendering method for reconstructing high-resolution geometry using radio frequency (RF) signals. GeRaF introduces three important techniques: lensless sampling, physics-based RF volumetric rendering, and a lensless alpha blending strategy. As a result, GeRaF reduces computational cost while maintaining the high-quality resolution of the baselines. The learnable part of the pipeline drops from cubic complexity to linear per iteration, slashing GPU memory.

**Questions:**

While I am unfamiliar with the area of this proposed work, I would like to raise a few questions.

1. Could you clarify whether the authors considered any baselines beyond the matched-filter (MF) approach?
2. In Figure 6, I think it is still hard to recognize the objects in the experimental results. Nevertheless, what specific advantages does GeRaF offer? For example, does it reconstruct certain features or regions particularly well compared to the baselines?

**Ethical Concerns:**

["NO or VERY MINOR ethics concerns only"]

**Final Justification:**

The authors addressed the reviewer’s concerns,

**Limitations:**

Yes, the authors clearly address the limitations and potential negative societal impact of GeRaF.

**Paper Formatting Concerns:**

None.

**Quality:**

3

**Strengths And Weaknesses:**

- Strengths
   - GeRaF extends NeRF-style learning to raw mmWave radar, enabling 3-D reconstruction even when the target is hidden behind thin occluders—something RGB/LiDAR can’t do.
   - Sharing density/transmittance across antennas and re-using a signal-tracing bank cuts the learnable complexity from $O(N_{\text{ant}}N_{\text{ray}}N_z)$ to $O(N_{\text{ray}}N_z)$.
   - Quantitative and visualization results outperform the matched filter (MF) method.

- Weaknesses
   - If a surface patch never reflects toward the radar in any scan, the network has no evidence and cannot hallucinate that geometry. Robust scanning strategies or priors are still needed.
   - GeRaF has three models: SDF network, Reflective network, and Signal power prediction. While training the three models, is there a risk that the exploration error could become disproportionately large in one of them, leading to instability? If so, how can this issue be mitigated?

---

> ### Author Rebuttal · Authors · 2025-07-30
>
> We would like to thank the reviewer for their comments and thought provoking questions. We especially appreciate the reviewer’s comments on our novel methods to boost the computational efficiency in bringing neural reconstruction to RF, which opens up reconstruction in cases where vision fails. Below, we will respond to the comments and concerns raised by the reviewer.
>
> __The reviewer comments on the robust scanning strategies required to fully reconstruct.__
>
> The reviewer is correct in the comment that robust scanning strategies or priors are still required in this work, however, we imagine future work will tackle the issue of adaptive scanning methods that make sure that we cover all viewpoints and/or will combine this work with generative models. However, we leave this issue to future work. In this work, we densely scan the object around 360 degrees to avoid missing patches.
>
> __The reviewer raises an interesting question regarding the stability of the three networks.__
>
> While training multiple coupled models (SDF network, Reflective network, Signal Power) as part of GeRaF, it is true that in some cases, instability may arise between the three networks. Asynchronous learning dynamics or gradient scale imbalance could cause one model (e.g., SDF network) to diverge or dominate the learning process. However, we take steps to mitigate potential instability by doing the following:
>
> 1. The Reflective Network is designed with: Sigmoid output in (0, 1), which ensures bounded influence on any downstream loss (e.g., ray attenuation, visibility). Initialization to constant 0.5, which acts as a neutral prior (neither fully reflective nor fully absorbing) and prevents early overconfidence or masking of supervision signals.
>
> 2. Signal Power Prediction is Not a Source of Instability. The signal power prediction model is a single parameter, not a high-capacity function approximator.
>
> 3. The SDF model is initialized to represent a sphere [1], providing  a strong geometric prior, anchoring early training around a meaningful shape and gradients that are spatially structured, preventing early overfitting to noise or partial signals.
> All three models interact through the end-to-end training pipeline and their contributions to the final signal are multiplicative or compositional, not adversarial. As a result, the reflective network scales the SDF-based surface responses, but does not compete with or override them and the signal power scalar normalizes the output, not drive dynamics.
>
> __The reviewer asks if any other baselines have been considered other than the Matched Filter.__
>
> In the realm of radar imaging and reconstruction, for general 3D reconstruction methods, there are generally three common deterministic methods: Matched Filter [3], Beamforming [4], and Range Migration Algorithm [5]. However, it is important to note that with the exception of Matched Filter, the other two algorithms rely on assumptions that lead to inaccuracies as compared to the Matched Filter. For non-deterministic methods, the majority of state-of-the-art work is done on far-field reconstruction, where the assumptions on radar physics are completely different, and where small objects are not considered. Thus, the Matched Filter approach is the most accurate radar imaging reconstruction method that we can compare to. However, we will explore adding more baselines such as Beamforming[4], Range Migration Algorithm [5] and NeRO[2] (for a comparison with vision).
>
> __The reviewer also raises concerns on the advantages of using our method.__
>
> Visually looking at the comparison of the Matched Filter mesh and the GeRaF mesh, we believe there is a clear improvement in our results over the current methods used in radar imaging. For example, the nose of the elephant completely disappears in the Matched Filter mesh, whereas we can see the nose in our reconstruction. Another clear difference is in the bottle, where the neck of the bottle is shown in the GeRaF reconstruction, but not the Matched Filter mesh, showing the ability of GeRaF to reconstruct more complex/thinner portions of the objects which the Matched Filter meshes cannot. These show a clear advantage of GeRaF over current state of the art.
> We would also like to specify that the Camera meshes shown in the results are all taken in line of sight (meaning the object is visible to the camera), however, many of our results also show the reconstruction in non-line-of-sight which the camera mesh completely fails (line of sight camera mesh is only shown for reference). Additionally, quantitatively GeRaF shows a large improvement over the Matched Filter result.
>
> We do acknowledge that since this is the first paper to introduce neural implicit reconstruction for RF signals, there are still limitations in terms of practicality and accuracy of results to overcome in this system.
>
> We again thank the reviewer for the thoughtful comments and questions, and hope that our answers clarify any concerns.
>
> [1] Atzmon, Matan, and Yaron Lipman. "Sal: Sign agnostic learning of shapes from raw data." Proceedings of the IEEE/CVF conference on computer vision and pattern recognition. 2020.
> [2] Liu, Yuan, et al. "Nero: Neural geometry and brdf reconstruction of reflective objects from multiview images." ACM Transactions on Graphics (ToG) 42.4 (2023): 1-22.
> [3] Novak, Leslie M., Gregory J. Owirka, and Christine M. Netishen. "Radar target identification using spatial matched filters." Pattern Recognition 27.4 (1994): 607-617.
>
> [4] Harter, Marlene, Andreas Ziroff, and Thomas Zwick. "Three-dimensional radar imaging by digital beamforming." 2011 8th European Radar Conference. IEEE, 2011.
>
> [5] Zhu, Rongqiang, et al. "Range migration algorithm for near-field MIMO-SAR imaging." IEEE Geoscience and Remote Sensing Letters 14.12 (2017): 2280-2284

---

> ### Comment · Reviewer_U8dg · 2025-08-05
>
> Thanks to the authors for the detailed rebuttals. All of the reviewer’s concerns have been addressed. Since the reviewer is not an expert in this research area, the reviewer will keep the current score, raise the confidence to 2 when providing the Final Justification, and continue the discussion during the reviewer–AC period.

---

> > ### Author Response · Authors · 2025-08-05
> >
> > We would like to sincerely thank you for your thoughtful review and feedback. Your comments are invaluable to us in improving the paper. Thank you once again!

---

### Official Review · Reviewer_1FNB · 2025-07-03

**Clarity:** 3
**Significance:** 3
**Originality:** 4
**Rating:** 4
**Confidence:** 3

**Summary:**

This paper presents a novel method for mesh reconstruction using radio frequency (RF) information, addressing occlusion challenges. The authors propose a neural rendering framework incorporating a Matched Filter (MF) and lensless sampling to model physics-based RF volume rendering. Qualitative and quantitative results demonstrate the method’s ability to reconstruct objects reasonably well, even under unseen lighting conditions.

**Questions:**

The paper states that the method works best for metallic objects. If the occluding object itself (e.g., the box) is metallic, would this introduce additional interference or artifacts in the reconstruction? Could the authors clarify how the system handles such cases?

**Ethical Concerns:**

["NO or VERY MINOR ethics concerns only"]

**Final Justification:**

All of my concerns about this paper have been well addressed. This cutting-edge work can inspire more ideas and research in the community, and I would recommend this paper to be accepted.

**Limitations:**

Yes

**Quality:**

3

**Strengths And Weaknesses:**

[Strengths]

- This paper proposes the first RF-based object-level mesh reconstruction method with occlusion handling, unlike prior RF works focusing on large-scale environments and novel view synthesis.
- The designed network is reasonable to me: The use of MLPs for SDF and reflectance networks, combined with MF image supervision and lensless sampling, efficiently reduces computational overhead.

[Weaknesses]

- While the authors use iPhone-scanned meshes as ground truth, the visualizations reveal noticeable inaccuracies in these reference meshes. This raises concerns about the reliability of both qualitative and quantitative evaluations, as the baseline itself may be flawed.
- The reconstructed mesh from MF images and GeRaF both lack detail, and the 32-hour training time per object raises concerns about scalability. The method’s real-world applicability is unclear. Suspicious package detection may be used by other efficient methods.

---

> ### Author Rebuttal · Authors · 2025-07-30
>
> We would like to thank the reviewer for their thoughtful comments and questions. We especially appreciate the reviewer’s comments on the novelty of the work as being the first work to propose RF based object mesh reconstruction. Below, we will answer the concerns and comments brought up by the reviewer point by point.
>
> __The reviewer raises concerns of noticeable inaccuracies in the reference meshes.__
>
> These inaccuracies are due to the fact that for specular objects, a line of sight reconstruction with vision is still a difficult problem. However, we believe the current baseline reference is a good comparison because it provides a general shape. It’s worth noting that in general RF reconstruction is much lower resolution, and hence as a first order, our goal is to reconstruct the high-level shape even if the minute details are not as accurate.
>
> That being said, the reviewer raises a good point, and so we have rescanned the objects after painting them with a matte paint for the objects with very noticeable inaccuracies (bunny, elephant, bottle) and the results along with the updated quantitative results will be included in the revised paper. The changes in quantitative results are shown in the table below:
> | Object   | Method         | F1 Score | Chamfer Distance (cm) |
> |----------|----------------|----------|------------------------|
> | Elephant | Matched Filter | 0.59 –> 0.65     |0.39 –> 0.25          |
> |          | GeRaF          |0.84 –> 0.79     |0.13 –> 0.14                  |
> | Bottle   | Matched Filter |0.1 –> 0.44     |1.30 –> 0.35             |
> |          | GeRaF          |0.88 –> 0.97     |0.09 –> 0.05                   |
> | Bunny    | Matched Filter |0.49 –> 0.56     |0.40 –> 0.03            |
> |          | GeRaF          |0.79 –> 0.81     |0.1 –> 0.01                   |
> | Elephant* | Matched Filter | 0.47 –> 0.42     |0.60 –> 0.52          |
> |          | GeRaF          |0.58 –> 0.52     |0.46 –> 0.46                  |
>
>
> __The reviewer also raises concerns regarding the scalability and use cases of this work.__
>
> We acknowledge that the training time is long, however, this is the first work in this area to tackle the issue of high resolution 3D mesh reconstruction using RF signals, which can open up many more applications than relying on visual models. While the current system as is, may not be the most efficient method for suspicious package detection or quality assurance, we believe that future work can push to lessen the gap to make GeRaF both scalable and more accurate.
>
> __The reviewer asks an interesting question regarding if the occluding object is metal.__
>
> This is a limitation due to the radio frequency (RF) characteristics, since the RF signal itself would not be able to penetrate most metal surfaces. This means anything behind a thick metal surface would be effectively invisible. However, in other cases where the occluding surface does not reflect or absorb all of the RF signal (e.g. thin aluminum), then this loss in propagation is modeled by the incoming transmit power to the rendered scene.
>
> Again we would like to thank the reviewer for their time in reviewing our paper and hope our answers clarify their uncertainties.

---

> > ### Comment · Reviewer_1FNB · 2025-08-07
> >
> > Thank you for the effort in the rebuttal. All of my concerns about this paper have been well addressed. This cutting-edge work can inspire more ideas and research in the community, and I would keep my original score.

---

> > > ### Author Response · Authors · 2025-08-08
> > >
> > > Thank you to the reviewer once again for the time put into thoroughly evaluating our work, and helping improve our paper.

---

### Decision · Program_Chairs · 2025-09-17

**Decision:**

Accept (spotlight)

**Comment:**

Radio frequency (RF) signals offer the possibility to see through occlusions, such as cardbord boxes (see Figure 1), without causing harm as it is the case for X-Ray waves. This paper resembles the first work that shows how 3D reconstruction can be performed from RF signals for object-level meshes. This is of very high practical relevance (e.g. intrusion-free damage or threat detection) and likely to spur many follow-up works.

The remaining concerns have been resolved and all reviewers vote for accepting the paper. Due to the postive reviews and the strong novel contribution, the AC recommends the paper for acceptance.